# PD-L1/PD-1 Axis in Glioblastoma Multiforme

**DOI:** 10.3390/ijms20215347

**Published:** 2019-10-28

**Authors:** Jakub Litak, Marek Mazurek, Cezary Grochowski, Piotr Kamieniak, Jacek Roliński

**Affiliations:** 1Department of Immunology, Medical University of Lublin, Jaczewskiego 8, 20-954 Lublin, Poland; jakub.litak@gmail.com (J.L.); jacek.rolinski@gmail.com (J.R.); 2Department of Neurosurgery and Pediatric Neurosurgery, Medical University of Lublin, Jaczewskiego 8, 20-954 Lublin, Poland; marekmazurek@hotmail.com (M.M.); pkamieniak@poczta.onet.pl (P.K.); 3Department of Anatomy, Medical University of Lublin, Jaczewskiego 4, 20-090 Lublin, Poland

**Keywords:** PD1, PD1 ligand, glioblastoma multiforme, GBM

## Abstract

Glioblastoma (GBM) is the most popular primary central nervous system cancer and has an extremely expansive course. Aggressive tumor growth correlates with short median overall survival (OS) oscillating between 14 and 17 months. The survival rate of patients in a three-year follow up oscillates around 10%. The interaction of the proteins programmed death-1 (PD-1) and programmed cell death ligand (PD-L1) creates an immunoregulatory axis promoting invasion of glioblastoma multiforme cells in the brain tissue. The PD-1 pathway maintains immunological homeostasis and protects against autoimmunity. PD-L1 expression on glioblastoma surface promotes PD-1 receptor activation in microglia, resulting in the negative regulation of T cell responses. Glioblastoma multiforme cells induce PD-L1 secretion by activation of various receptors such as toll like receptor (TLR), epidermal growth factor receptor (EGFR), interferon alpha receptor (IFNAR), interferon-gamma receptor (IFNGR). Binding of the PD-1 ligand to the PD-1 receptor activates the protein tyrosine phosphatase SHP-2, which dephosphorylates Zap 70, and this inhibits T cell proliferation and downregulates lymphocyte cytotoxic activity. Relevant studies demonstrated that the expression of PD-L1 in glioma correlates with WHO grading and could be considered as a tumor biomarker. Studies in preclinical GBM mouse models confirmed the safety and efficiency of monoclonal antibodies targeting the PD-1/PD-L1 axis. Satisfactory results such as significant regression of tumor mass and longer animal survival time were observed. Monoclonal antibodies inhibiting PD-1 and PD-L1 are being tested in clinical trials concerning patients with recurrent glioblastoma multiforme.

## 1. Introduction

Glioblastoma (GBM) is the most common primary cancer in the central nervous system and has an extremely expansive course [1]. The standard approach to glioma treatment consists in the most extensive as possible surgical resection and in adjuvant radiation strengthened by temozolomide (TMZ) administration [2]. The median overall survival (OS) oscillates between 14 and 17 months [3,4]. The survival rate of patients in a three-year follow up oscillates around 10% [5]. GBM resistance to typical therapies requires verification. Glioblastoma cells interact with the surrounding environment, creating forceful interactions among heterogenous cell groups, various chemokines with cytogenetic effects, and extracellular proteins stimulating tumorigenesis, uncontrolled multifocal expansion, and immunological evasion [6].

The proteins programmed death-1 (PD-1) and programmed cell death ligand (PD-L1) interplay, creating an immunoregulatory axis promoting invasion of glioblastoma multiforme cells in the brain tissue [7]. Physiologically, the main function of PD-1 is to restrain T-cell anti-tumor activity and amplify Tregs activation, which limit T-cell reaction and protect against hyper immunity. The PD-L1/PD-1 axis maintains immunological homeostasis and protects against autoimmunity [8].

## 2. Central Nervous System Lymphatic System

The central nervous system (CNS) was believed an “immune privileged” organ. This idea was based on anatomical features limiting the immune response, such as the presence of the blood–brain barrier (BBB) and the lack of a typical lymphatic system [9,10]. Several studies challenged this dogma. In 2015, the discovery of a lymphatic CNS system was made. Studies revealed that the CNS has a specific connection with deep cervical lymph nodes. Antigens and T cells are transferred through cerebrospinal fluid-filled channels. APC cell (microglia macrophages and dendritic cells) activate T cells which can reach the CNS though the perivascular space and recognize neoplastic cells, a process which is under the control of the PD-1/ PD-L1 axis. The PD-1 receptor is expressed especially on activated T cells. Overexpression of PD-L1 in glioblastoma multiforme cells and microglia promotes effective binding of PD-L1 to PD-1, causing a negative regulation of the immune response. This negative regulation affects mainly T cell responses [11,12].

The dysfunction of the BBB commonly noted in patient with glioblastoma multiforme and the concomitant inflammation enhances the CNS interplay with the lymphatic system [13].

Glioma cells excrete suppressive cytokines to stop regional responses. Prostaglandins, especially prostaglandin E2 (PGE-2), and interleukin-10 act directly on nearby intercellular matrix; TGF-β has a similar restrictive effect on regional immune responses. Glioblastoma multiforme upregulates the superficial expression of immune checkpoint molecules such as PD-L1/PD1 which promotes immune inhibition [14].

According to evidence, GBM has the ability to eliminate CD4+ T cell and Th1-related cytotoxic response from the neoplastic tissue. GBM may turn the immune response into chronic inflammation mediated by Th17 cells, creating convenient conditions for tumor expansion [15,16].

## 3. PD-L1 and PD-1 Structure

PD-L1 is encoded by the gene *PDCDL1*, localized on the 9th chromosome in p24.1 position. PD-L1 is also described as CD274 and B7-H1. This ligand was discovered and described in 1999 by Dong et al. as a member of the B7 protein family [17]. Seven exons encode the full-length protein PD-L1, consisting of 290 amino acids (40 skDa). PD-L1 as a type-I transmembrane complex of proteins includes single IgV and IgC domains on the external part, a transmembrane domain with hydrophobic properties, and a 30-amino acid cytoplasmatic tail as a signal transducer [18].

PD-L1 activity depends on binding to the PD-1 receptor encoded by the *PDCD1* gene. It is information transcribed from the second chromosome and consists of 288 amino acids (50–55 kDa). It contains an IgV domain in the extracellular domain and transmembrane region [19].

The intracellular region forms a tail composed of a tyrosine-based switch motif (ITSM)–inhibitory motif. This receptor was described by Ishida et al., who used subtractive hybridization to identify genes regulating programmed cell death [20]. PD-L1 is expressed and excreted by neoplastic cells, APCs, lymphocytes B, and parenchymal cells. It induces T-cell apoptosis or anergy and modulates inflammation in situ [21,22].

The binding of PD-1 to the corresponding PD-1 receptor activates the protein tyrosine phosphatase SHP-2, which dephosphorylates Zap 70 (Figure 1). This process T cells proliferation and downregulates lymphocyte cytotoxic activity [23].

Studies revealed that glioma cells are the principal expressors of PD-1 ligands [24]. The presence of PD-L1 was revealed in glioblastoma lines and biopsies in 2003 by Wintterle et al. [25]. Relevant studies demonstrated that the presence of PD-1L in glioma cells correlates with WHO grading and could be considered as a biomarker for glioma cells [26].

## 4. PD-1 Ligand Expression—Role of TLR Activation

Toll-like receptors (TLRs) and agonists of receptors induce the immune response, activating many pathways, and cooperate with various antigens. TLRs are a conserved family of 10 receptors (TLR1–10) taking part in pattern recognition [27]. Agonists of this group of receptors used to be called pathogen-associated molecular patterns (PAMPs). Their binding to specific TLRs initiates an immune response [28,29]. Studies revealed that TLRs are endogenously expressed in glioma cells. The fact that TLR2, TLR4, and TLR9 activated by agonists promote tumor expansion and proliferation complicates the role of TLRs in the antitumor response [30].

TLR agonists as microbial antigens, activating TLR receptors to initiate precise immunological activities. Agonists of TLR that have been studied include lipopeptides (TLR2, TLR6, and TLR1 agonists), lipopolysaccharides (TLR4 agonist), LPS, flagellin (agonists of TLR5), single-stranded DNA (TLR8 agonist and TLR7), double-stranded (ds)DNA (agonist of TLR3), and the DNA CpG motif (agonist of TLR9). Later studies showed that autocrine molecules released from dead and stressed cells such as heat shock proteins (HSP, for TLR4 and TLR2) and high-mobility group box 1 proteins (HMGB1, for TLR4 and TLR2) are also significant agonists. Many of them appear in glioma environment, causing tumor induced-activity of TLRs [31,32].

In GBM cells, constitutive elevated expression of αB-crystallin, HSP27, HSP73, HSP72, and HSP90 was reported in vivo and in vitro as a result of endogenous induction. HSP–peptide complexes (HSPPCs) are able to interfere with various superficial receptors such as CD36, CD91, CD40, CD14, TLR2, TLR4 [33,34,35].

TLR activation in glioma cell results in signaling through two main pathways, one of which is myeloid differentiation factor 88-independent (MyD88-independent), and the other is myeloid differentiation factor 88-dependent (MyD88-dependent) (Figure 2). The MyD88-dependent signaling cascade (MyD88/TRAF6/MEK/ERK) promotes early activation of cytokine transcription through NF-κB, supporting inflammatory processes and cytosolic enzyme and chemokine activity, and starts PD-L1 gene transcription. The MyD88-independent pathway leads to the late activation of NF-κB and interferon regulatory factors (IRF), which control the expression of type I IFNs and the activation of many gene promoters. Excreted Type I IFNs influence PD-L1 overexpression through IFNAR signaling activation, proving an indirect effect of the independent pathway [36,37,38].

## 5. EGFR Pathway

EGFR (epidermal growth factor receptor) was firstly discovered during the search for tyrosine kinases [39]. Previous observations reported that the sequence of EGFR was similar to that of ErbB tyrosine kinase, known as a prooncogenic protein. Since then, EGFR has been detected as overexpressed in a number of tumors of epithelial origin [40]. Its inappropriate activation leads to altered apoptosis and/or intensified angiogenesis, necrosis, proliferation as well as treatment refractoriness, suggesting a relationship between receptor dysregulation and the biology of various cancers. Amplification of the EGFR gene and its mutations are also found in lung, prostate, and breast cancers [41].

EGFR activation requires ligand binding. Binding of TGFα or EGF to EGFR promotes the activation of selected pathways: Ras/RAF/MAPK, PI3K/Akt-1, and signal transducer and transcription-activating (STAT) pathways [42,43,44,45].

Incidental mutation of EGFR results in the recruitment of this downstream signaling pathway which normally promotes immune escape after typical EGF binding. EGFRvIII is a modified type of EGFR receptor, transmitting immunological signals constantly in the absence of EGF. EGFRvIII expression rate is about 11% in GBM.

Recent studies revealed PD-1 activation and downstream signaling of EGFR were interdependent and both present in pulmonary tumors [46,47]. The activity of EGFR increases the expression of PD-L1. A mouse model of glioblastoma demonstrated EGFR protooncogenic interplay supporting PTEN deletion [48]. Removal of PTEN promoted the overexpression of PD-L1, indicating a strong relationship with EGFR. PTEN loss resulted in increased PD-L1 expression. Some researchers presented a direct correlation between the rate of PD-L1 expression and cancer expansion, indicating that an immune-resistant phenotype may promote significant tumor progression. Tumor aggressiveness and PD-1L expression are directly related to downstream signaling through the PI(3)K/Akt/mTORS6K1 pathway in neoplastic cells [49,50,51,52,53].

## 6. Angiogenesis in Glioblastomas

Features of glioblastomas are a high propensity to invade the surrounding tissues and extensive angiogenesis [54]. They are characterized by a pathological vascular system with a dense network of winding vessels of different diameters, increased permeability, and with abnormally thickened basement membranes. These properties are often the cause of relapses and therapy failures because they hinder tumor access to cytotoxic chemotherapy [55,56,57].

Angiogenesis caused by the presence of a tumor is defined as the formation of new blood vessels based on existing ones, which can be visualized using ultra-high-field magnetic resonance angiography [58]. The mechanism of this process is very complex. In GBM, the development of pathological vasculature is based on changes in adhesion molecules on the endothelial surface, the presence of an erratic basement membrane, a decrease in pericyte coverage, and a weaker binding of astrocytes to blood vessels [59]. A common element for the large invasiveness and angiogenesis of glioblastomas is the loss of tumor cell boundaries with the surrounding microenvironment consisting of brain tissue, subarachnoid space, and blood vessels [60].

Angiogenesis occurs under the influence of various mechanisms, which we divide into hypoxia-dependent and hypoxia-independent. Hypoxia has been proven to inactivate prolyl hydroxylases. This increases the level of hypoxia-inducible factor-1α (HIF-1α), causing the release of vascular endothelial growth factor (VEGF) by glioma cells affected by hypoxia. The task of VEGF is to counteract oxygen deficiency by inducing vascular network growth [61,62,63].

In spite of this, hypoxia may not be the only trigger factor for angiogenesis. This applies particularly to the initial stages of carcinogenesis, when cell hypoxia is not yet severe. Regulation of VEGF secretion also takes place by a mechanism independent of hypoxia, involving dysregulated activation of mitogenic and survival pathways. This applies to phosphatidylinositol 3-kinase and Ras/mitogen-activated protein kinase [63,64].

The PD-L1/PD-1 axis may also have roles in regulating the level of VEGF mediating angiogenesis. In studies conducted by Shin et al., a positive correlation between PDL1 levels and VEGF expression was shown in 197 patients with renal cell carcinoma [65,66]. Similar observations were noted by Song et al. The study was conducted in 64 patients with primary glioma. Also, they showed that PD-L1 expression levels correlated well with VEGF levels. However, not all studies agreed on this point. Research by Joseph et al. indicates the existence of a negative relationship between VEGF-related genes and the activity of the PD-L1/PD1 axis [3]. A thorough understanding of the nature of this relationship requires further work.

EGF is another molecule with similar effects. It promotes tumor angiogenesis as well as an increase in cell motility and invasiveness. This is done by phosphorylating the EGFR receptor and activating further signaling pathways [67,68]. It has been shown that both EGFR and its mutated form of EGFRvIII promote GBM angiogenesis [69,70,71]. To allow vasculature growth and tumor invasion, it is necessary to reduce the integrity of the tumor microenvironment. The pathophysiology of this process is based on EGFR inducing an increase in the activity of some proteases, thereby causing the breakdown of collagen, fibronectin, and other basement membrane structures. The result is a loss of microenvironment integrity, an increase in tumor cell invasion into surrounding tissues and vessels, and an increase in angiogenesis. The proteases whose activity is influenced by EGFR include, among others:Serine ProteasesMatrix Metalloproteases (MMPs)Cysteine Proteases [72].

Serine proteases are a large group of proteases which include tissue plasminogen activator (tPA) and urokinase plasminogen activator (uPA) with its receptor (uPAR). Their activation affects the degradation of the extracellular space by tumors, promoting their invasiveness. It has been shown that, the PAI-1 molecule mediates the regulation of the uPA/uPAR pathway in astrocytoma cells. Its expression is promoted by an increase in EGF and inflammatory cytokines. This results in greater uPA/uPAR activity, thus increasing tumor invasiveness [71].

Upregulation of uPA and uPAR is also achieved through the expression of the EGFR ligand transforming growth factor (TGFα) [73]. Studies confirmed that in malignant gliomas, both PAI-1 and EGFR activity correlate with higher tumor malignancy and worse prognosis for patients [74,75].

EGF also has a potential impact on another group of proteases, i.e., MMPs. They are a large group of enzymes whose effect is based on the regulation of metabolism and cell morphogenesis. Disruption of their function also affects the loss of cell integrity. It has been shown that the activity of certain matrix metalloproteases, such as MMP-2 (gelatinase A), MMP-9 (gelatinase B), MMP-14 (matrix metalloproteinase 14) is related to the pathophysiology of high-grade gliomas. It correlates with the severity of tumor invasion and its degree of malignancy [76,77,78,79].

Another group of proteases are cysteine proteases, examples of which are cathepsins. These include, among others, cathepsin B, involved in the degradation of the tumor microenvironment. Its expression (mRNA level) has been shown to correlate with EGFR expression level [80]. It was also confirmed that its presence is increased in gliomas [81]. The action of cathepsin B may be bidirectional. It promotes the degradation of the integrity of the environment of the tumor directly or through intermediate mediators such as uPA, MMPs, or plasmin [82].

In addition to VEGF and EGF, factors that contribute to angiogenesis also include other molecules: platelet-derived growth factor (PDGF), fibroblast growth factor (FGF), hepatocyte growth factor, ephrins, angiopoietins, integrins, and interleukin-8 [83,84].

## 7. INF Receptor

Interferons were first described in the 1950s as molecules interfering with viral replication. Signaling pathways from their receptors through Janus kinase (JAK) and STAT have been clearly described. This main signaling pathway mediates interferon-induced gene expression [85,86,87,88]. Currently, IFN has gained attention because of its potential role in regulating PD-1L expression [89]. IFN type 1 (including α, β, and ω) binds to type 1 interferon receptor, which consists of two subunits: IFNAR 1 and IFNAR 2. Signal transduction via STAT 1–3 cascade and JAK1 and TYK2 then occurs. After activation, STAT1 forms the ISGF3 complex with interferon regulatory factor 9 (IRF9). This complex binds to ISRE sequences at the genomic level, controlling many interferon-induced genes [90,91]. Type I interferon also has the ability to phosphorylate other representatives of the STAT group: STAT3, STAT4, STAT5, and STAT6 [92].

Another consequence of activating the interferon signaling pathway is an increase in the activity of cytoplasmic GTPases, resulting in an antiviral effect. It can also lead to increased expression of myxovirus protein A (MxA) [93,94].

It has been shown that an increase in MxA mRNA accompanies a greater degree of tumor malignancy. The highest values were observed in the case of glioblastoma, but a good correlation was also observed in vivo in patients diagnosed with other gliomas. In this case, there was also a correlation with PD-L1 mRNA levels. MxA values can also be associated with the presence of nonclassical MHC molecules, including HLA-E and HLA-G, which promote glioma immune escape.

The existence of an autocrine signaling loop in which interferon participates has been proven. In this loop, INF-a and INF-b (IFN I types) trigger intracellular signaling through IFNAR1/2.

*IFNAR1/2* gene silencing resulted in reduced expression of PD-L1 and MHC (class I and II). As a result, the immune cells were more susceptible to lysis. On the basis of this, it can be concluded that intracellular signals induced by the presence of interferon negatively regulate the anti-tumor immune response in gliomas [95].

The binding of type II interferon (IFN-γ) to its receptor (IFNGR) causes phosphorylation of JAK1 and JAK2 as well as the phosphorylation of STAT1. Activated dimers act after accumulation in the nucleus as transcription factors and connect to GAS elements that are located on most IFN-γ-induced genes. One of them is the *IRF*-1 gene [96,97,98]. It has been reported that cells treated with IFN-γ express PD-L1 on their surface [99,100].

Evidence indicates that the presence of PD-L1 on the cell surface is regulated by downstream transcription interferon regulatory factor 1 (IRF-1). STAT1 and STAT3 also play a role [101].

To sum up interferon pathways, researchers suggest that stimulation of the IFN-y pathway results in the upregulation of JAK2/STAT1/IRF1 and, as a consequence, increase the expression of PD-L1 as a characteristic signaling transmission element for type II interferon. Some studies noticed that IFN-y stimulated the STAT3, STAT2, and IRF9 axis which is more typical of type I interferon stimulation. This confirms a strong relation between the two molecules [102].

## 8. Immunotherapy with Anti PD-1/PD-L1

Monoclonal antibodies (mAbs) have produced a turning point in the immunotherapy of cancer. They block receptors on immune effector cells or their ligands on neoplastic cells and accessory cells (antigen-presenting cells, APCs). Understanding the role of PD-L1 and PD-1 in the pathogenesis of neoplasms has enabled the blocking of these mechanisms in oncological therapy. Currently, there are several drugs in clinical trials that could be used to treat, among others, lung cancer, ovarian and stomach cancer, renal cell carcinoma, melanoma, lymphoma, and leukemia [103,104,105,106,107].

In some countries, several mAbs have been approved by the relevant authorities for clinical use [108]. One of them is NIVOLUMAB, which is a human IgG4 anti-PD-1 monoclonal antibody. It works by blocking PD-1 molecules, modulating the immune response [103,104]. PEMBROLIZUMAB is another example. It is a humanized mAb that also works by blocking PD-1. Both FDA (U.S. Food and Drug Administration) and EMA (European Medicines Agency) have approved the clinical use of these drugs for the treatment of non-small-cell lung carcinoma in patients previously treated with chemotherapy. Nivolumab was also approved in Japan [108]. In addition to lung cancer, it has been shown to have a beneficial effect in the treatment of kidney cancer, melanoma, and Chodkin's lymphoma [103,104]. Some drugs use PD-L1 blocking as a mechanism. We include in this group ATEZOLIZUMAB, which is humanized mAb, as well as DURVALUMAB and AVELUMAB, which contain completely human sequences [106,107]. The use of PD-1 and PD-L1 immunotherapy is also investigated in the case of glioblastoma.

Preclinical GBM mouse model studies confirmed the safety and efficiency of monoclonal antibodies directed against PD-1 and PD-L1. The data obtained so far, such as significant regression of tumor mass and longer animal survival time, suggest their high anti-cancer potential [7,109,110].

Monoclonal antibodies as inhibitors of PD1 and PD-L1 are being tested in trials concerning patients with recurrent GBM (Table 1) [111,112,113,114,115,116,117,118,119,120].

A study performed by Reardon et al. in 2017, called CheckMate 143 trial, included 369 patients with glioblastoma who had their first relapse after radiation and TMZ therapy. Observations compared nivolumab efficiency administered every 2 weeks at a dose of 3 mg/kg with bevacizumab administered every 2 weeks at a dose of 10 mg/kg. The median overall survival in the group of patients taking nivolumab was 9.8 months and that in in the corresponding bevacizumab group was 10 months. The median duration of response in the nivolumab cohort was 11 months versus 5.3 months in the bevacizumab cohort. Overall response rates in the nivolumab group were 8%, while in the bevacizumab group, they were 23% [121].

Another drug from this group, Durvalumab, was tested in an open label, phase 2 trial where patients were grouped into five GBM subgroups, performed by Reardon et al. in 2017. The study design involved a monotherapy with Durvalumab at a dose of 10 mg/kg. Partial results revealed a partial response of 13,3% in bevacizumab-naïve recurrent GBM patients and a stable response in 46,7% of patients in this cohort (*n* = 30). Four patients were progression-free at 12 months [119] (Table 1).

Although these trials did not confirm an extension of OS in the bevacizumab group, the introduction of inhibitor checkpoints in the therapy of glioblastoma may confer different benefits. Further clinical trials are required to create the best protocols for inhibitors of immune check points in GBM patients.

It should be emphasized that the best possible clinical outcome is usually obtained by the simultaneous use of different types of therapy based on a variety of anti-cancer mechanisms.

A combined therapy with PD-L1 inhibitor and MAPK or PI3K inhibitors showed promising results in the case of a mouse model [122,123]. Analogous observations have not yet been made for glioblastoma; multiple-targeted immunotherapy may be an opportunity to improve the outcomes of treatment for patients with this type of neoplasm.

Despite promising research results, anti PD-1 and anti PD-L1 therapy cannot be used for all patients. It has been shown that among GBM patients, expression of programmed death-ligand 1 in tumor cells is found in 61–88% of patients [124,125]. However, the presence of PD-L1 is not a dummy variable. Severity is determined by the expression level of PD-L1 within a certain range [126] and also depends on the GBM histological type [124,125].

PD-L1 expression has been shown to influence the clinical response of patients receiving anti-PD-1 and anti-PD-L1 immunotherapy. Patients whose tumors had PD-L1 expression showed a more pronounced response to anti-PD-1 treatment compared to those whose PD-L1 levels were low. However, these studies concerned tumors other than gliomas (lung, prostate, colorectal, melanoma, and renal cell carcinoma) [127].

Similar findings were obtained in other studies which examined the correlation between the level of PD-L1 in tumor cells, lymphocytes PD1, and clinical response. In the case of PD-1 level, the degree of this relationship was poor [128].

However, accurate knowledge of this correlation requires further research. Many patients in the PD-L1-expressing group are unreactive to blockade of relevant checkpoints. On the other hand, it has been shown that in the case of lung cancer, a part of the study group, negative for PD-L1, showed a good clinical response [129,130].

## 9. Summary 

GBM is highly resistant to standard therapies and thus is a challenging disease. Short OS and aggressive intercourse remain uncontrollable when using the typical therapeutic approach. The PD-1L/PD-1 immune check point axis is considered an interesting target for immunotherapy. Many pathways activating PD-L1/PD-1-dependent glioma cell immunoescape were clearly described. Blockade of the interaction between PD-1 and PD-L1 may re-establish proper immunity against GBM. Recent clinical studies on PD-1L/PD-1 inhibitors have presented effective responses. Although trials do not confirm critical extension of OS, these inhibitors may confer different benefits if combined with standard therapies. Further clinical observations are required to create the best therapeutic protocols that can bring a new hope for patients.

## Figures and Tables

**Figure 1 ijms-20-05347-f001:**
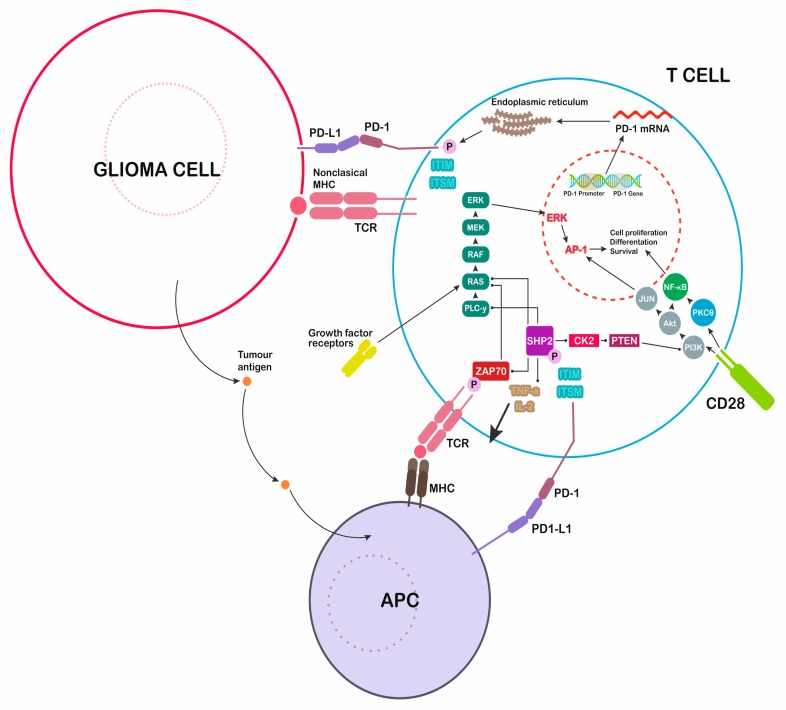
Immunological modulation induced by glioblastoma multiforme (GBM) cells. Secretion of programmed cell death ligand 1 (PD-L1) inhibits the immune attack, blocking T cells responses. PD-1: programmed death-1, major histocompatibility complex: MHC, T-cell receptor: TCR, antigen-presenting cell: APC.

**Figure 2 ijms-20-05347-f002:**
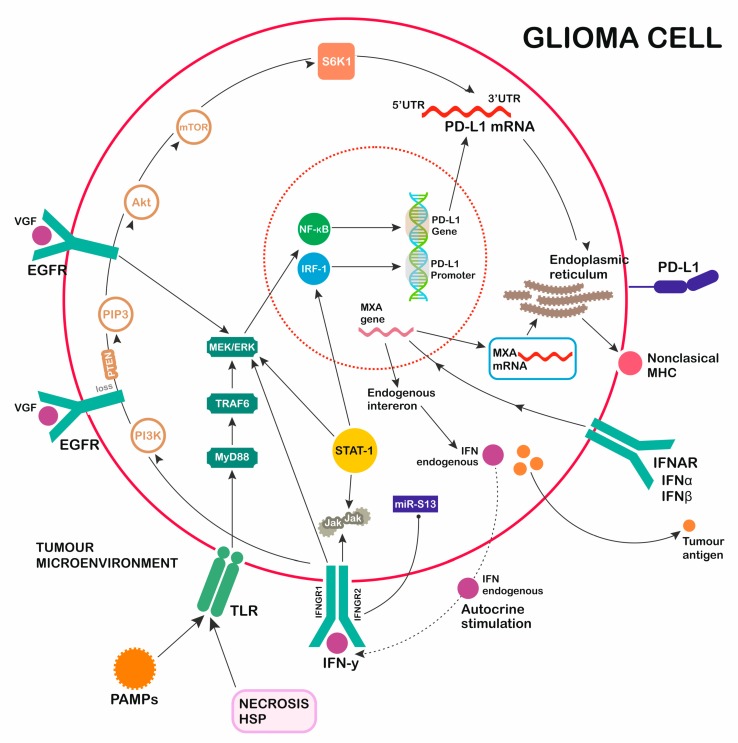
GBM induction of PD-L1 secretion. Multiple activation pathways (TLR, EGFR, IFNAR, IFNGR) promoting PD-L1 expression. 1. Toll-like receptors (TLR) pathway: pathogen-associated molecular patterns (PAMPs), NECROSIS, heat shock proteins (HSP) as activators of TLR myeloid differentiation factor 88 (MyD88)-dependent pathway signaling through TRAF6/MEK/ERK/NF-κB. 2. Epidermal growth factor (EGFR) pathway: TGFα/EGF/VGF/MUTATION OF RECEPTOR as activators of EGFR pathway signaling through MEK/ERK (STAT-1)/NF-κB. 3. IFNAR pathway: interferon (IFN)alfa, IFNbeta as activators of IFNAR pathway signaling through MXA gene transcription, forming nonclassical MHC, promoting PD-1L transcription, and the induction of endogenous interferons. 4. IFNGR pathway: IFNgamma as an activator of IIFNGR pathway signaling through JAK/STAT-1/MEK/ERK/IRF-1 and PI3K/PIP3/Akt/mTOR/S6K1, with a regulatory function over transcribed PD-1L mRNA.

**Table 1 ijms-20-05347-t001:** I and phase II trials of monoclonal antibodies anti-PD-1/PD-L1.

Clinical trial Identification Number	Study Phase	Number of Patients	Monoclonal Antibody
NCT02829931	I	26	Nivolumab
NCT02313272	I	46	Pembrolizumab, bevacizumab
NCT02529072	I	66	Nivolumab
NCT02658981	I	68	Anti-LAG-3, urelumab, nivolumab
NCT02526017	I	280	nivolumab
NCT03233152	I	6	Ipilimumab, nivolumab
NCT02937844	I	20	Anti-PD-L1 CSR T cells
NCT03058289	I/II	60	anti-PD-1 antibody
NCT02327078	I/II	291	Nivolumab, epacadostat
NCT02311582	I/II	52	MK-3475, MRI-guided laser ablation
NCT02866747	I/II	62	Durvalumab
NCT02798406	II	48	pembrolizumab
NCT02335918	II	205	Varlilumab, nivolumab
NCT02968940	II	43	Avelumab
NCT02794883	II	36	Durvalumab, tremelimumab
NCT02336165	II	159	bevacizumab
NCT02337491	II	82	Pembrolizumab, bevacizumab
NCT03014804	II	30	Nivolumab
NCT02550249	II	29	Nivolumab
NCT02852655	Pilot	30	Pembrolizumab

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
