# Peer review of "PD-L1/PD-1 Axis in Glioblastoma Multiforme"

_ijms, 2019, doi:10.3390/ijms20215347_

Round 1

Reviewer 1 Report

The review article by Jakub Litak et al described the PDL1/PD1 axis in gliomblastoma multiforme (GBM). This review article has interesting aspects but there are a number of issues that need to be addressed. For these reasons, I don’t recommend publishing this manuscript with current version. After the minor revision/correction, this manuscript will be further strengthened. Please find my minor comments enclosed below.

Minor comments.

PD-L1 and PD-1 should be spelled at the first time appearance and doesn’t need to be spelled afterwards. In the throughout manuscript, PD-L1 is written as PD-L1, PD-1L, and PD-1 ligand. Please unify PD-L1 in the entire manuscript thoroughly. Figure legends was not described in detail. Please describe in detail. Section 4 subtitle is “PD-1 ligand expression-role of TLR activation”. However, authors described the TLR in detail but didn’t clearly describe how TLR activation is related with PD-1 ligand expression. Please describe the mechanism and relation between PD-1 Ligand expression and TLR activation in detail. In the section 6, there is no explanation to link PDL1-PD1 axis in angiogenesis in GBM. Describe more details how PDL1-PD1 axis is related in angiogenesis in GBM. If the summary or conclusion is added, the article will be improved further. In several places, there are typo and grammatical errors. Please revise the entire manuscript thoroughly for typos and grammatical errors.

Author Response

We would like to thank the reviewers for the corrections and suggestions. We believe that those suggestions greatly improve the quality of the manuscript.

PD-L1 and PD-1 should be spelled at the first time appearance and doesn’t need to be spelled afterwards. - Corrected In the throughout manuscript, PD-L1 is written as PD-L1, PD-1L, and PD-1 ligand. Please unify PD-L1 in the entire manuscript thoroughly. - Corrected. We described the processes presented in the figure. We thoroughly described the the relation between PD-L1 and TLR activation We provided detailed description of the relation between angiogenesis and GBM. We added summary section We corrected all the typos. 

Reviewer 2 Report

A comprehensive review by Litak and colleagues is an accurate and wide-ranged source of current data dealing with the application of monoclonal antibodies acting as Pd-1L/PD-1 axis inhibitors in the treatment of glioblastoma multiforme.  The topic is very timely and of potential interest for the readers, especially from the clinical viewpoint. However, several minor points have to be revised as follows:

Line 15. Please write “microglia” instead of microglias. Line 24. It should be “antibodies”. Line 29. The sentence is a repetition of the first sentence of the abstract. Furthermore, I suggest to write: “GBM is the most common primary central nervous system cancer…. Line 30. It should be “contains”. Line 33 It should be “oscillates’. Line 36.”tumorigenesis” is a correct term Line 51. Use “recognize” Line 53. There should be “glioblastoma”. Line 65. Change into “convenient”. Line 74. It should be “transcribed”. Line 106. Term “flagellin (agonist of TLR5)” appears twice. Please delete one item Line 165. “hypoxia-inducible” sounds better. Line 184. Is should be “Metaloproteases” Line 205. “cathepsin” is a correct name. Line 214. It should be “transducer”. Line 243. Write: “evidences indicate…” Line 246. It should probably be: “stimulation” Line 250. “stimulation”. Section 315-319 does not really fit to the scope of the article, it is poorly documented and should be removed as well as appropriate reference.

Authors use two different forms of antibody names interchangeably: capitalized or not. Please apply one form constantly in the whole text and tables.

Author Response

We would like to thank the reviewers for the corrections and suggestions. We believe that those suggestions greatly improve the quality of the manuscript.

We corrected all the typos and mistakes, that the reviewer suggested.

We removed the section 315-319.

We applied one form constantly in the text.